# Synthesis and Hypoglycemic Effect of Insulin from the Venom of Sea Anemone *Exaiptasia diaphana*

**DOI:** 10.3390/md22030111

**Published:** 2024-02-27

**Authors:** Qiqi Guo, Tianle Tang, Jingyue Lu, Meiling Huang, Junqing Zhang, Linlin Ma, Bingmiao Gao

**Affiliations:** 1Engineering Research Center of Tropical Medicine Innovation and Transformation, Ministry of Education, Hainan Key Laboratory for Research and Development of Tropical Herbs, School of Pharmacy, Hainan Medical University, Haikou 571199, China; qiqiguo@hainmc.edu.cn (Q.G.); tianle_tang@hainmc.edu.cn (T.T.); lujingyue6426@163.com (J.L.); mgling213@163.com (M.H.); jqzhang2011@163.com (J.Z.); 2Griffith Institute for Drug Discovery (GRIDD), School of Environment and Science, Griffith University, Nathan, QLD 4111, Australia

**Keywords:** sea anemone, insulin-like peptides, *Exaiptasia diaphana*, synthesis, hypoglycemic

## Abstract

Sea anemone venom, abundant in protein and peptide toxins, serves primarily for predatory defense and competition. This study delves into the insulin-like peptides (ILPs) present in sea anemones, particularly focusing on their role in potentially inducing hypoglycemic shock in prey. We identified five distinct ILPs in *Exaiptasia diaphana*, exhibiting varied sequences. Among these, ILP-Ap04 was successfully synthesized using solid phase peptide synthesis (SPPS) to evaluate its hypoglycemic activity. When tested in zebrafish, ILP-Ap04 significantly reduced blood glucose levels in a model of diabetes induced by streptozotocin (STZ) and glucose, concurrently affecting the normal locomotor behavior of zebrafish larvae. Furthermore, molecular docking studies revealed ILP-Ap04’s unique interaction with the human insulin receptor, characterized by a detailed hydrogen-bonding network, which supports a unique mechanism for its hypoglycemic effects. Our findings suggest that sea anemones have evolved sophisticated strategies to activate insulin receptors in vertebrates, providing innovative insights into the design of novel drugs for the treatment of diabetes.

## 1. Introduction

Sea anemones (Cnidaria: Anthozoa: Hexacorallia: Actiniaria) represent one of the most ancient groups of venomous animals, equipped with a complex arsenal of peptide toxins that serve for defense, predation, and intro- and interspecific competition [1,2,3,4,5]. These venomous peptides, with their rich pharmacological potential and biological diversity, constitute a valuable resource for marine drug discovery [5]. Among the myriad of bioactive compounds, peptide neurotoxins have emerged as a rapidly developing field, gaining attention for their application in neuroscience and potential as novel therapeutic agents. For example, Dalazaride (formerly known as ShK-186), derived from sea anemone *Stichodactyla helianthus*, was found to selectively block K_V_1.3 channel, and it represents a significant area of study for its therapeutic potential in autoimmune and inflammatory diseases, such as multiple sclerosis, rheumatoid arthritis and psoriasis [6,7].

Sea anemone venom is a complex mixture of peptides, proteins, and non-protein compounds (such as purines, quaternary ammonium compounds, and biogenic amines) [4]. Peptides and proteins are the primary focus of current research. These peptides, defined by their amino acid count (less than 80 for peptides), are characterized into different families based on cysteine patterns, such as the κ_1.3_-SHTX-She1a (ShK), Kunitz-type, β-Defensin and epidermal growth factor-like (EGF-like) [8].

Beyond its traditional role in metabolism, growth, and development, insulin and insulin-like peptides (ILPs) also play crucial roles in reproduction, longevity, and stress response [9,10,11,12,13]. While the functions of insulin and ILPs in vertebrates have been extensively studied, their presence and role in invertebrates, especially in venomous species, are beginning to unfold, revealing a spectrum of biological functions. For example, ILPs in garter snakes affect their temperature sensitivity, genetic background, and growth variation [14]. While in *Drosophila melanogaster*, they can mediate antiviral immunity, acting as an important component of the insect diapause regulation pathway [13,15,16]. An ILP isolated from the Asian malaria mosquito, *Anopheles stephensi*, acts as a steroidogenic gonadotropin across diverse mosquito taxa [17,18]. This cross-species diversity in ILP functions underscores the potential for novel research into their mechanisms and applications.

Many discoveries have highlighted the presence and intriguing roles of ILPs in marine organisms, sparking considerable interest in their functions and therapeutic potential. Recently, Gao et al. [19] elucidated the diversity and evolutionary relationship of venom insulins derived from cone snails, reporting 38 different insulin sequences in the venom of 18 cone snail species. Notably, research into venom-derived insulin, such as that from the fish-hunting cone snail *Conus geographus*, has revealed specialized insulins capable of inducing hypoglycemic “insulin shock” to immobilize prey, pointing to a new avenue for understanding ILP functions and applications in invertebrates [20,21,22,23,24].

Similar to cone snails, sea anemones may also produce ILPs as prey-hunting weapons. Indeed, five insulin sequences were found in the sea anemone *Exaiptasia diaphana* through transcriptomic studies [25], which have the characteristic two-chain (A and B) and a typical three-disulfide-bond structure of human insulin. To further understand the biological functions of these sea anemone ILPs, this study focuses on ILP-Ap04, one of the five insulin sequences identified in *Exaiptasia diaphana*. Distinct from its counterparts, ILP-Ap04 exhibits significant specificity in its amino acid sequences and 3D structures. Therefore, it was synthesized through solid-phase peptide synthesis (SPPS) to investigate its hypoglycemic activity in zebrafish models of hyperglycemia and behavioral assays. Molecular docking further elucidated its action mechanism, aiming to contribute to the development of innovative diabetes treatments and enhance our understanding of marine-derived pharmaceuticals.

## 2. Results

### 2.1. The Five Novel ILPs Identified in the Sea Anemone Exaiptasia diaphana

In our previous research, five ILPs were identified in the sea anemone *Exaiptasia diaphana* through transcriptome analysis [25], each displaying the characteristic two-chain (A and B chains) structure and the typical three-disulfide-bond configuration found in human insulin (Figure 1A). To further investigate the potential hypoglycemic activity of these novel ILPs from sea anemone *Exaiptasia diaphana*, this study focuses on ILP-Ap04, selected for its notable sequence specificity and structural characteristics. ILP-Ap04 is composed of an A-chain with 28 amino acids and a B-chain with 20 amino acids, linked together by three disulfide bonds. The homologous modeling results show that ILP-Ap01, ILP-Ap02, and ILP-Ap03 share structural similarities with human insulin, while ILP-Ap04 and ILP-Ap05 exhibit significant structural deviations (Figure 1B), indicating that these ILPs may have distinct hypoglycemic mechanisms.

### 2.2. RP-HPLC Analysis and LC-MS Identification of ILP-Ap04

The synthesis of sea anemone ILPs, characterized by three disulfide bonds, employs an oxidation method. ILP-Ap04, composed of 48 amino acids (AAs) and featuring three-disulfide-bond linkages, exhibits structural similarities to human insulin and other ILPs. The linear peptides were assembled through the SPPS protocol with two free Cys, two Cys modified with Acm, and two Cys modified with tBu. The theoretical molecular weights of linear peptide A chain and B chain are 3494.94 Da and 2403.68 Da, respectively. Analysis via liquid chromatograph mass spectrometer (LC-MS) and reversed-phase high-performance liquid chromatography (RP-HPLC) confirmed the actual molecular weights of the linear peptide A chain and B chain as 3493.66 Da and 2404.17 Da, respectively, indicating successful synthesis. Subsequent purification of peptides A and B using preparative RP-HPLC achieved over 95% purity for both peptides, with retention times of 34.595 min and 33.765 min, respectively (Figure 2A).

The initial oxidative folding step allows the linear peptide A chain and B chain to connect through a pair of free cysteine, resulting in a combined molecular weight of 5895.79 Da, with a retention time of 34.823 min. This step led to a minor decrease in total molecular weight by 2.03 Da, indicating the precise removal of two hydrogen atoms (Figure 2B). The second oxidation phase removes two Acm protective groups (144 Da), establishing the second disulfide bond, and reducing the molecular weights to 5751.74 Da, which is consistent with the expected difference of 144.05 Da, the retention time extended to 36.453 min (Figure 2C). The final stage involves heating in a water bath heating with a large amount of TFA, which removes the tBu protective group (114 Da) and forms the third disulfide bond. The resulting peptide chain’s molecular weight adjusted to 5637.71 Da, aligning with the removal of two tBu protective groups after two oxidation steps, and exhibited a retention time of 29.475 min (Figure 2D).

Figure 2E illustrates the dynamic changes in retention time throughout the oxidative folding process on RP-HPLC. Notably, compared to the linear peptides, the oxidative folding stages sequentially delayed the retention time by about half a minute and two minutes, respectively, while the final step accelerates the retention time by about four minutes, showcasing the methodical progression toward the fully folded ILP-Ap04.

### 2.3. CD Spectroscopic Analysis of ILP-Ap04 Oxidative Products

Circular dichroism (CD) spectroscopy is an important technique for studying protein structures, with a particular emphasis on elucidating secondary structures and conformational changes during biological processes [26]. In this study, we compared and assessed the secondary structure conformations of the human insulin, In-1, In-2, and In-3 using CD spectroscopic analysis (Figure 3). Figure 3A shows the amino acid arrangement of ILP-Ap04 and the pattern of disulfide-bond formation during the oxidation process. The CD spectroscopic results revealed that In-1, In-2, and In-3 all have secondary structures similar to that of human insulin, characterized by a distinct minimum at the wavelength of 210 nm, indicative of typical insulin disulfide-bond configurations (Figure 3B). In addition, the quantified changes in the secondary structure composition through the oxidative folding process (Figure 3C). As the three-step oxidative folding progresses, the content of α-helix gradually decreases, while the content of strands, turns, and unordered regions steadily increases. Specifically, the content of the α-helix decreased from 0.694 after the initial oxidation folding step to 0.444, nearing the α-helix proportion of human insulin (0.354). Concurrently, the content of the strands increased from 0.020 to 0.156, approaching the strand content of human insulin (0.195). These changes signify that through oxidative folding, the secondary structure of ILP-Ap04 progressively approaches that of human insulin, demonstrating the refinement of ILP-Ap04’s protein structure towards a conformation similar to human insulin.

### 2.4. ILP-Ap04 Reduces Blood Glucose in a Zebrafish Model of Diabetes

To assess the potential hypoglycemic effects of ILP-Ap04, we employed two zebrafish models of diabetes induced by streptozotocin (STZ) and glucose, respectively [27,28,29]. After intraperitoneal injection of STZ, the blood glucose levels of zebrafish significantly increased from 3.9 ± 3.1 mmol/L to 9.8 ± 7.6 mmol/L (*n* > 5). Blood glucose levels surged rapidly within two hours post-STZ injection and remained elevated for an extended period (Figure 4A). The administration of human insulin and ILP-Ap04 at a dosage of 65 ng/g significantly reduced the blood glucose levels in zebrafish. The most pronounced reduction was observed at 30 min post-administration, with blood glucose reaching 5.0 ± 2.7 mmol/L and 4.7 ± 2.2 mmol/L (*n* = 8) for human insulin and ILP-Ap04, respectively (Figure 4B). This outcome aligns with findings from Helena Safavi Hemami et al., where human insulin at the same dosage decreased blood glucose to 92.0 ± 17.35 mg/dL (5.11 ± 0.97 mmol/L), corroborating the results of this study [24,30]. An examination of the hypoglycemic effects of varying concentrations of human insulin and ILP-Ap04 revealed their half-maximal effective concentration (EC_50_) to be 56.66 ng/g and 50.17 ng/g, respectively (Figure 4C). In conclusion, ILP-Ap04 showed strong hypoglycemic activity in the STZ-induced zebrafish model of diabetes, and its activity was comparable to that of human insulin.

In the glucose-induced model of diabetes, zebrafish displayed a marked increase in blood glucose levels one hour after injection (14.3 ± 4.9 mmol/L, *n* = 8), while those treated with human insulin and ILP-Ap04 showed significantly lower glucose levels compared to the glucose group (Figure 4D). These findings from both the STZ and glucose-induced diabetes models indicate that the ILP-AP04 may be able to bind and activate the insulin receptors in fish, supporting their biological role in inducing insulin shock.

### 2.5. ILP-Ap04 Interferes with the Locomotor Behavior of Zebrafish Larvae

Locomotion activity tests were performed to investigate the influence of ILP-Ap04 exposure for 30 min on the behavior of zebrafish larvae (Figure 5). As the dose of ILP-Ap04 was increased, a progressive decrease in the distance covered was observed, with the most pronounced reduction occurring at the highest tested concentration of 1.0 μM (Figure 5A,B). Although no statistical significance was detected when the zebrafish larvae were treated with 0.25 μM ILP-Ap04, the total distance traveled by the 1 μM ILP-Ap04-treated zebrafish larvae was significantly lower than that of the control (Figure 5B). In contrast, high doses (1.0 μM) of human insulin did not exhibit a comparable effect, indicating that human insulin does not significantly impact the locomotor behavior of zebrafish larvae. In addition, the total distance covered by the zebrafish larvae showed minimal variation across different durations, exhibiting a stable pattern without notable increases or decreases over time (Figure 5C).

### 2.6. Prediction of Binding Patterns of ILP-Ap04 on Human Insulin Receptor

Given the notable similarities between ILP-Ap04 and human insulin, along with its hypoglycemic effects observed in zebrafish diabetes models, it is plausible to suggest that ILP-Ap04 might also interact with the human insulin receptor. To investigate this possibility, we utilized the ZDOCK and RDOCK algorithms to simulate the potential interaction between ILP-Ap04 and the human insulin receptor (PDB 6PXV). The ZDOCK score was 24.38 and the E_RDOCK score was 43.52. A higher ZDOCK score and a lower E_RDOCK score indicate that the docking result of this pose is better and closer to the true docking conformation.

The analysis of the hydrophobic interaction within the complex formed by the human insulin receptor and ILP-Ap04 showed that the Polar Contact Surface Area at its interface was 131.35 Å2 (receptor) and 66.53 Å2 (ligand), while the Nonpolar Contact Surface Area was 207.46 Å2 (receptor) and 14.68 Å2 (ligand), respectively. Critical amino acids such as Asn15, Phe39, Lys40, Leu62, Phe64, Arg65, Phe88, Phe89, Tyr91, Val94, Phe96, Glu97, Arg118, Glu120, Lys121, Tyr144, Leu147, and Glu154 in the human insulin receptor, along with Arg52, Ser53, Arg68, and Arg69 in ILP-Ap04, exhibited hydrophobic properties, facilitating a robust hydrophobic interaction at their interface.

Hydrogen bonding is an important force in molecular interactions, playing a crucial role in the stability of the complex. In order to further understand the mechanism by which ILP-Ap04 activates the human insulin receptor signaling pathway, a macromolecular docking simulation was carried out to ascertain whether there is a direct interaction between ILP-Ap04 and human insulin receptors (Figure 6A). The results highlight that ILP-Ap04 engages the pocket of human insulin receptors through three hydrogen bonds (Figure 6B), involving Arg-42 (2.8 Å), Glu-124 (2.6 Å), and Asn-123 (2.6 Å). These residues may play an important role in maintaining the structural stability of the receptor-ILP-Ap04 complex.

## 3. Discussion

The intriguing complexity of marine animal venom, composed of a mixture of various bioactive peptide toxins that target multiple receptors, underscores not only the evolutionary adaptations of these creatures but also unveils a promising frontier for biomedical research and medical therapies [8,31,32]. The identification of anti-diabetic ILPs in marine organisms such as cone snails and sea anemones exemplify this new horizon [33,34]. Such ILPs could represent exciting exploitable scaffolds for future drug discovery in diabetes, as well as provide tools to allow for a better understanding of cell signaling pathways linked to insulin secretion and metabolism, which has potential biotechnology applications. The discovery of ILPs in the venom of fish-hunting cone snails, which facilitate prey capture by rapidly inducing hypoglycemic shock, marks a significant advancement. One representative example, Con-Ins G1 derived from *Conus geographus*, is the smallest known ILP found in nature [30]. The study of Con-Ins G1 revealed its unique monomeric structure and potent interaction with the human insulin receptor, highlighting its potential for therapeutic application [35]. Therefore, these ILPs will provide a set of solutions to potentially solve a long-standing problem of designing truly monomeric, fast-acting insulin analogs for the treatment of diabetes [20,24,35].

Moreover, the rapid growth of multi-omics research has unveiled a wealth of ILPs from the genomes, proteomes, and transcriptomes of sea anemones, including *Nematostella vectensis* [36,37], *Exaiptasia diaphana* [25], and so on. However, sea anemone ILPs have not been characterized functionally. Unlike other animals, cnidarians lack a circulatory and central nervous system, which adds a layer of complexity in understanding the physiological roles of ILPs in these organisms. Nevertheless, some progress has been made. Pascual et al. [38] demonstrated that aqueous crude extracts from the sea anemones *Bunodosoma granuliferum* and *Bartholomea annulata* inhibit porcine dipeptidyl peptidase IV (DPPIV). Inhibition of this enzyme results in a reduction in glucagon and blood glucose levels, and may therefore be useful as an alternative treatment for diabetes.

Recently, our investigation into ILPs derived from sea anemone *Exaiptasia diaphana* has led to the identification of five novel ILPs with structural similarities to human insulin [25]. With a similar approach, several ILPs were identified in the tentacle transcriptomes of *Oulactis* sp. [39,40]. Bioactivity studies of IlO1_i1 were conducted on human insulin and insulin-like growth factor receptors, and on voltage-gated potassium, sodium, and calcium channels [41]. IlO1_i1 did not bind to the insulin or insulin-like growth factor receptors, but showed weak activity against K_V_1.2, 1.3, 3.1, and 11.1 (hERG) channels, as well as Na_V_1.4 channels [41].

In this study, ILP-Ap04 was used as a representative for synthesis, oxidative folding, and activity testing among the five ILPs. The complex molecular architecture of ILP-Ap04, which is composed of 48 amino acids, forming two chains and three disulfide bonds, along with the poor physical and chemical properties of individual chains, has made the synthesis of this peptide highly challenging [42]. Employing a series of strategies, we successfully achieved selective formation of disulfide bonds and directional oxidative folding of the two chains (Figure 2). The CD spectroscopy measurements indicate that the formation of disulfide bonds during the three-step oxidative folding process has an impact on the structure (Figure 3). In comparison, the CD spectra of IlO1_i1 did not display the characteristic helical content of insulin and were largely disordered, even though it has canonical insulin disulfide connectivity [40]. Therefore, our findings indicate that the correct formation of disulfide bonds is crucial for mimicking the bioactive conformation of human insulin, underscoring the intricate balance between structure and function in protein biochemistry.

When tested in zebrafish, ILP-Ap04, like human insulin, significantly lowers blood glucose in the STZ and glucose-induced model of diabetes (Figure 4). Our results are consistent with the findings of Helena Safavi Hemami et al., where human insulin at the same dosage decreased blood glucose [24,30]. Therefore, the structural and functional aspects of ILP-Ap04 could be compatible with human insulin receptors. This compatibility suggests a potential for cross-species efficacy, advocating that sea anemone-derived insulin could broaden the spectrum of insulin therapies. The differential efficacy of ILP-Ap04 in STZ versus glucose-induced diabetes models (Figure 4C) may provide valuable insights into the mechanisms of insulin receptor interaction and insulin resistance. Understanding how ILP-Ap04 modulates blood glucose levels could lead to the development of insulins with improved receptor affinity or reduced propensity for inducing insulin resistance. In addition, the distinct hypoglycemic effects observed in different diabetes models highlight the possibility of tailoring insulin therapies to specific types of diabetes or patient needs. Insulins like ILP-Ap04 could be optimized for longer shelf-life, reduced risk of eliciting immune responses, and for particular diabetic conditions, improving personalized medicine approaches in diabetes care.

Additionally, the observed behavioral effects of ILP-Ap04 on zebrafish offer a novel perspective on the physiological impact of venom-derived insulin in a natural context. The administration of ILP-Ap04 to zebrafish larvae caused significantly shorter swimming distances, while human insulin did not show a similar effect (Figure 5). This intriguing observation suggests that, compared to human insulin, ILP-Ap04 has a stronger impact on the physiological functions of zebrafish due to its hypoglycemic effect.

Our molecular docking studies reveal a unique interaction mechanism of ILP-Ap04 with the insulin receptor, diverging from the traditional binding paradigms observed in human insulin. Similar to IlO1_i1, ILP-Ap04 also lacks the key FFY motif at the end of the B-chain, which is crucial for human insulin to engage with the high-affinity binding site on the insulin receptor [30,35]. The molecular docking results indicate that the ILP-Ap04 interacts with insulin receptors through hydrogen bonding and hydrophobic interaction, respectively (Figure 6). This novel interaction suggests alternative pathways for insulin receptor activation, providing a foundation for the design of new insulin analogs that could offer therapeutic advantages over existing treatments.

## 4. Materials and Methods

### 4.1. Screening of Peptide Sequence

Our research group previously reported an article on the transcriptome sequencing of sea anemone *Exaiptasia diaphana*, which discovered five types of ILPs (ILP-Ap01-ILP-Ap05) [25]. Five ILPs were compared with human insulin sequences in the database using NCBI-BLAST (https://blast.ncbi.nlm.nih.gov/, accessed on 1 May 2023). B-chain and A-chain cleavage sites were determined as per documented cleavage positions for human insulin (Lys-Arg, Arg-Arg) [43]. Using the structure of human insulin (PDB 6PXV) as a template, SWISS-MODEL (https://swissmodel.expasy.org/, accessed on 10 May 2023) was used to predict the 3D structure of five ILPs and then perform a comparative analysis. According to the above results, ILP-Ap04 was selected, and the next step of synthesis and activity research was carried out.

### 4.2. Synthesis of Linear Peptides

ILP-Ap04 is a new ILP identified from sea anemone *Exaiptasia diphana* through transcriptome sequencing, which contains 48 amino acids, including the A-chain (amino acids A1–A28) and B-chain (amino acids B1–B20). For the A-chain peptide, S-protection was afforded using tBu (Cys8,13), and Acm (Cys22) while for the B-chain, Acm (Cys15) was used. The linear peptide ILP-Ap04 was synthesized using SPPS [44]. ILP-Ap04 was synthesized using automated microwave synthesis conditions on a CEM Liberty PRIME 2.0 system at 0.1 mmol scale using the one-pot coupling/deprotection methodology [45]. The resin peptide was cut with a cutting solution (TFA:TIPS:DODT:H_2_O = 92.5%:2.5%:2.5%:2.5%) at 40 °C. After suction filtration, the crude peptide was washed with ether precooled using dry ice and centrifuged three times, then subsequently dissolved in 100% H_2_O.

### 4.3. Oxidative Folding

Selective disulfide formation was used to ensure that the native disulfide isomer of ILP was obtained. For the linear peptide ILP-Ap04, containing six cysteine residues can form three disulfide bonds (A9–B3, A22–B15, and A8–A13), with Acm (Cys22 in the A-chain peptide and Cys15 in the B-chain peptide) and tBu (Cys8 and Cys13 in the A-chain peptide) providing S-protection.

To form the first disulfide bond [Cys9 in the A-chain peptide and Cys3 in the B-chain peptide], lyophilized linear peptide ILP-Ap04 was dissolved in a mixed solution of methanol/H_2_O and diluted 10 times with acetic acid. The iodine solution was then added drop-by-drop. The mixture was continuously stirred for 10 h to form the first disulfide bond [Cys9 in the A-chain peptide and Cys3 in the B-chain peptide], and the product was quenched with ascorbic acid, and then analyzed via RP-HPLC and LC-MS. The first pair of cysteines was oxidized through this procedure, while other cysteines were protected from oxidation using the protecting groups.

To form the second disulfide bond [Cys22(Acm) in the A-chain peptide and Cys15(Acm) in the B-chain peptide], an equal volume of methanol hydrochloric acid and excessive iodine solution were added to the first-step product, and the mixture was stirred continuously for 1.5 h. The reaction was terminated by adding ascorbic acid to the peptide solution after the formation of the second disulfide bond [Cys22(Acm) in the A-chain peptide and Cys15(Acm) in the B-chain peptide]. The product was purified and analyzed via RP-HPLC and identified using LC-MS.

To form the third disulfide bond [Cys8(tBu) and Cys13(tBu) in the A-chain peptide], the oxidation product was dissolved in TFA/DMSO/anisole solution, stirred for 45 min at room temperature, then mixed with the TFA/DMSO/anisole solution again through continuous stirring in a 70 °C water bath for 3 h. The final product was purified and analyzed using RP-HPLC, identified through LC-MS, and then stored as powder via freeze-drying.

### 4.4. RP-HPLC Purification, Analysis and LC-MS Identification

The peptide solution was purified using P3500 preparative liquid chromatograph (Elite Analytical Instruments Co., Ltd., Dalian, China) on an Aglient InfinityLab Pursuit XRs 5 C18 column (21.2 × 250 mm). Mobile phase A and mobile phase B are H_2_O and acetonitrile, respectively. Elution was performed with a linear gradient of 5-50% solvent B over 45 min at a flow rate of 5 mL/min with absorbance monitored at 220 nm. The fractions were analyzed through Ultimate 3000 Ultra High-Performance Liquid Chromatography (Thermo Fisher Scientific, Waltham, MA, USA) on a Waters XBridge^®^ peptide BEH C18 (4.6 × 250 mm) using the same elution gradient. The molecular weight of the peptide was determined using a Triple TOF4600 mass spectrometer (Applied Biosystems, Waltham, MA, USA).

### 4.5. ILP-Ap04 Reduce Blood Glucose in the STZ and Glucose-Induced Model of Diabetes

#### 4.5.1. Zebrafish

The experimental zebrafish was purchased from the flower market in Haikou City, Hainan Province. A healthy, lively, and standardized AB wild-type blue zebrafish aged around 4~5 months, weighing around 200 mg, was fed twice a day (8:30 and 16:30) under standard environmental conditions. After 7 days of cultivation, the experiment began. The water temperature was controlled at 28.0~30.0 °C throughout the entire process, with a pH value of 7 ± 0.5, and the light cycle was 12 h of light and 12 h of dark cycling. The embryos were collected after natural pairwise mating (3~12 months). Then, the embryos and larvae were raised in an incubator system maintained at 28.5 °C in embryo medium.

#### 4.5.2. The STZ-Induced Model of Diabetes

In order to determine whether venom insulins are capable of lowering blood glucose in fish via activation of the insulin receptor, the ILP-Ap04 was tested in the STZ-induced model of diabetes in zebrafish [24,30]. According to the weight of zebrafish 100 mg/μL administration, intraperitoneal injection of 1.5 g/kg STZ to induce hyperglycemia in adult zebrafish [46]. Blood glucose levels were measured within 12 h (once per hour) to establish a hyperglycemic model. After 2 h of injection of STZ, 65ng/g of human insulin, ILP-Ap04, and physiological saline were injected intravenously for hypoglycemic activity. This concentration has previously been shown to be effective when using human insulin and venom insulin Con-Ins G1 [30]. After 15 min, 30 min, 45 min, 60 min, 120 min, and 180 min, measure blood glucose in mmol/L using a Bayer Contour meter. The data were analyzed in GraphPad Prism (8.0.1).

#### 4.5.3. The Glucose-Induced Model of Diabetes

After the last feeding of adult zebrafish, they were starved for 72 h, and then ice water bath anesthesia was used to suck up the water on the surface of the fish. It was then put on the ME Analytical balance to measure its weight, and the intraperitoneal injection method was used to administer the drug (100 mg/μL). This comprised an intraperitoneal injection of 0.1 M glucose solution, copolymer of 0.1M glucose and human insulin (1.69 IU/kg), and copolymer of 0.1 M glucose and ILP-Ap04 (65 ng/g), respectively. Blood samples were collected at 0 h, 1 h, 2 h, and 3 h after intraperitoneal injection, and blood glucose was measured with a Bayer Contour meter, in mmol/L. The data were analyzed in GraphPad Prism (8.0.1).

#### 4.5.4. Locomotion Behavior of Zebrafish Larvae Test

Zebrafish larvae at 5-day post-fertilization (5-dpf) were exposed to different concentrations (1.0 μM, 0.5 μM, and 0.25 μM) of ILP-Ap04 for 30 min, and human insulin (1.0 μM) was used as a positive control. Then, larvae were transferred to a 96-well plate (1 fish/well). Zebrafish locomotion recording was quickly initiated, and locomotion was measured for 20 min using the zebrafish tracking system (Viewpoint Life Sciences, Montreal, QC, Canada). Ten sessions (2 min each) were recorded for each zebrafish larva. The total distance traveled by each zebrafish was simultaneously calculated across sessions.

### 4.6. Circular Dichroism Spectroscopy Studies

CD spectra were recorded on a MOS-500 CD spectrometer from 400 to 190 nm with a 1.0 nm step size using a 1.0 s response time and 1.0 nm bandwidth in a quartz cuvette with a 0.2 cm path length. Human insulin and ILP-Ap04 were dissolved in a methanol solution at a concentration of 0.2 mg/mL. To correct for background, the spectrum of buffer alone was subtracted from each sample spectrum. The machine units collected—θ in millidegrees, were converted to mean residue ellipticity (MRE), [θ] (degrees·cm^2^dmol^−1^ residue^−1^) [41].

Helical content was calculated using the CDSSTR algorithm for deconvolution against the reference protein database set SMP180 [47,48]. This program is available on the DICROWEB website (http://dichroweb.cryst.bbk.ac.uk/html/home.shtml, accessed on 20 May 2023).

### 4.7. Molecular Docking Study

Human insulin (PDB 6PXV) [49] was used as the docking receptor, and the ILP-Ap04 in this study was used as the docking ligand. The Protein Preparation Wizard in Discovery Studio 2023 was used to optimize the structure of receptors and ligands [50], including removing crystalline water, processing disulfide bonds, processing metal ions, etc., and adding terminal hydrogen atoms to protein molecules according to expected temperature and pH. Subsequently, the ZDOCK module in Discovery Studio 2023 was used to predict the complex structure of human insulin receptor and ILP-Ap04. Set the Euler angle step size of the ligand direction for rotational sampling to 6, RMSD Cutoff to 6.0, Interface Cutoff to 9.0, Maximum Number of Clusters to 60, and Top Poses to 2000. All other parameters were calculated using default values. Then, the ZRANK method was used to reorder the ZDOCK docking scores. Finally, based on CHARMM polar H, the RDOCK method was used to re-optimize and score the ZDOCK results, selecting the conformation with the lowest E_RDOCK scores for analysis [51]. The human insulin receptor and ILP-Ap04 from the binding modes optimized by RDOCK were compared with the conformations of existing proteins in the crystal library through a superimpose program, and the conformation with RMSD < 3 Å was selected.

The Analyze Protein Interface module was applied to calculate the solvent-accessible surface area (SAS) of the human insulin receptor and ILP-Ap04, and analyze the key amino acid residues at the interface between the human insulin receptor and ILP-Ap04. The Calculate Interaction Energy module was used to calculate the interaction energy between key amino acid residues at the interface between the human insulin receptor and ILP-Ap04. By analyzing the interactions between key amino acid residues at the interface between human insulin receptor and ILP-Ap04, the binding sites of the human insulin receptor and ILP-Ap04 were predicted. Finally, the visualization combination mode was downloaded using PyMOL 2.6.

### 4.8. Statistical Analysis

Statistical analyses were performed using GraphPad Prism software (version 8.0; GraphPad Software, Inc., San Diego, CA, USA). Data are expressed as means ± standard deviation (SD). Differences between means were considered statistically significant. * Means significant difference (*p* < 0.05), ** means extremely significant difference (*p* < 0.01), and *** means extremely significant difference (*p* < 0.001).

## 5. Conclusions

The ILPs found in sea anemone venom represent a fascinating example of nature’s ingenuity, serving a critical role in prey capture through the induction of rapid hypoglycemic shock. This study has spotlighted ILP-Ap04, derived from *Exaiptasia diaphana*, marking it as the inaugural instance of confirmed hypoglycemic activity linked to sea anemone venom insulin. Notably, ILP-Ap04 is characterized by the absence of the C-terminal segment of the B-chain, a deviation from the typical insulin structure. Molecular docking studies have elucidated how the absence of the FFY motif is compensated by the presence of two asparagine and proline residues, showcasing a novel mechanism of receptor interaction. These findings underscore the unique properties of ILP-Ap04, which not only expand our understanding of venom-derived peptides, but also open new avenues for therapeutic innovation, highlighting the potential of sea anemone venom insulin as a template for novel drug design in diabetes management. 

## Figures and Tables

**Figure 1 marinedrugs-22-00111-f001:**
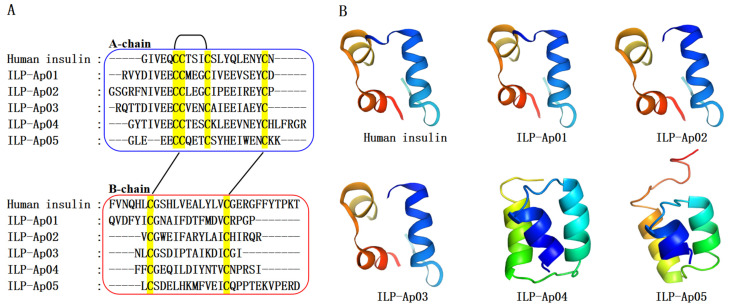
Sequence alignment and structural comparison of human insulin and the five newly identified ILPs from *Exaiptasia diaphana*. (**A**) Sequence alignment of A-chain and B-chain of the human insulin and ILP-Ap01~05. The conserved cysteines are highlighted with a yellow background. The A-chain and B-chain are displayed in blue and red boxes, respectively. The disulfide bond is displayed with black line segments and curves, connecting cysteine; (**B**) comparison of the 3D structure of human insulin (PDB 6PXV) and predicted structures of ILP-Ap01~05 based on homology modeling.

**Figure 2 marinedrugs-22-00111-f002:**
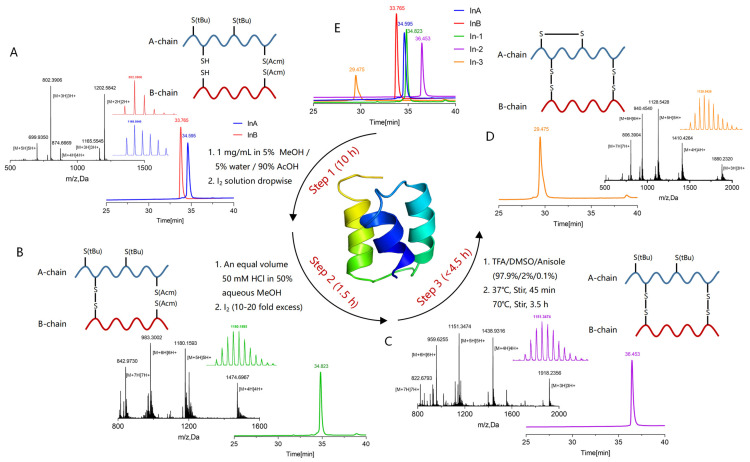
The purification and identification of ILP-Ap04 through successive oxidative folding steps. (**A**) RP-HPLC and LC-MS analyses of linear peptides A and B chains, denoted as InA and InB, respectively; (**B**–**D**) RP-HPLC and LC-MS results of the first, second, and third steps of oxidative folding, respectively; (**E**) RP-HPLC comparison of the entire oxidative folding process for ILP-Ap04. The products resulting from each oxidative folding phase are labeled as In-1, In-2, and In-3, corresponding to the first, second, and third steps, respectively.

**Figure 3 marinedrugs-22-00111-f003:**
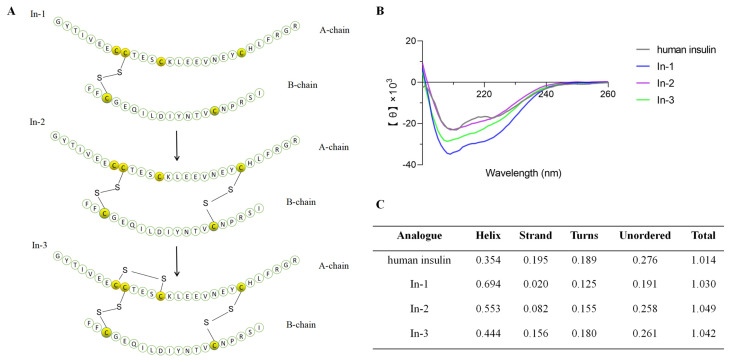
CD spectroscopic analysis of ILP-Ap04 oxidative products. (**A**) The amino acid arrangement of ILP-Ap04 oxidative products from the three oxidative folding phases (In-1, In-2, and In-3) and the disulfide-bond patterns formed during the oxidation process; (**B**) the CD spectroscopic analyses of human insulin and ILP-Ap04 oxidative products; (**C**) relative content of protein secondary structures.

**Figure 4 marinedrugs-22-00111-f004:**
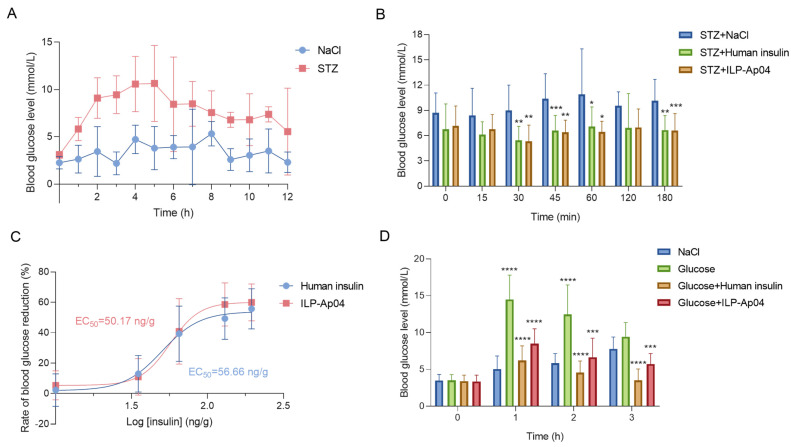
ILP-Ap04 demonstrates effective blood glucose reduction in STZ and glucose-induced diabetes models. (**A**) The STZ-induced model of diabetes; (**B**) hyperglycemia was effectively mitigated following the administration of 65 ng/g of either human insulin or ILP-Ap04; (**C**) effects of the administration of human insulin/ILP-Ap04 at various concentrations on the blood glucose levels of zebrafish; (**D**) the hypoglycemic effects of ILP-Ap04 tested in the glucose-induced model of diabetes. Comparisons were made between the glucose group and the NaCl group, as well as between the treatment groups and the glucose group, using One-way ANOVA. Data are presented as means ± SD (*n* > 5). * *p* < 0.05, ** *p* < 0.01, *** *p* < 0.001, **** *p* < 0.0001.

**Figure 5 marinedrugs-22-00111-f005:**
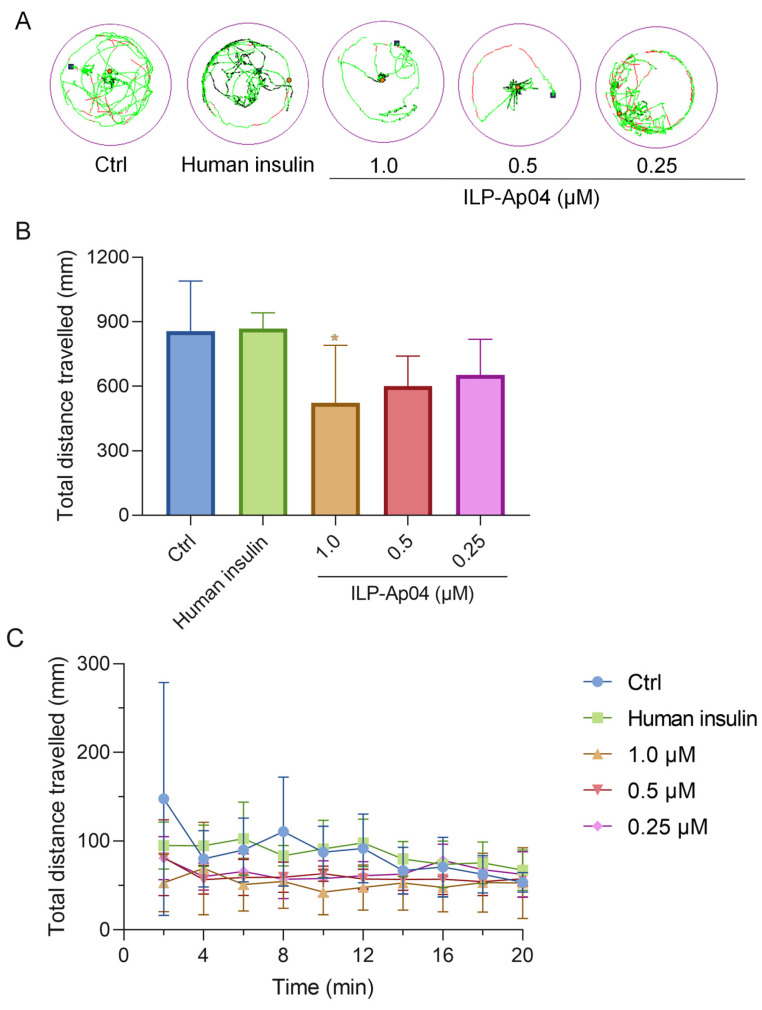
ILP-Ap04 interferes with the locomotor behavior of zebrafish larvae. (**A**) Representative patterns of locomotion behavior of zebrafish larvae. The swimming trajectory was recorded every 5 s and is represented by curves. The instantaneous velocity was detected and displayed in different colors (black, <2 mm/s; green, 2–8 mm/s; red, >8 mm/s); (**B**) the total distance traveled over 20 min. Comparisons were made between the treatment groups and the ctrl group, using One-way ANOVA; (**C**) changes in total distance travelled over 20 min. Data are expressed as means ± SD (*n* > 5), * *p* < 0.05.

**Figure 6 marinedrugs-22-00111-f006:**
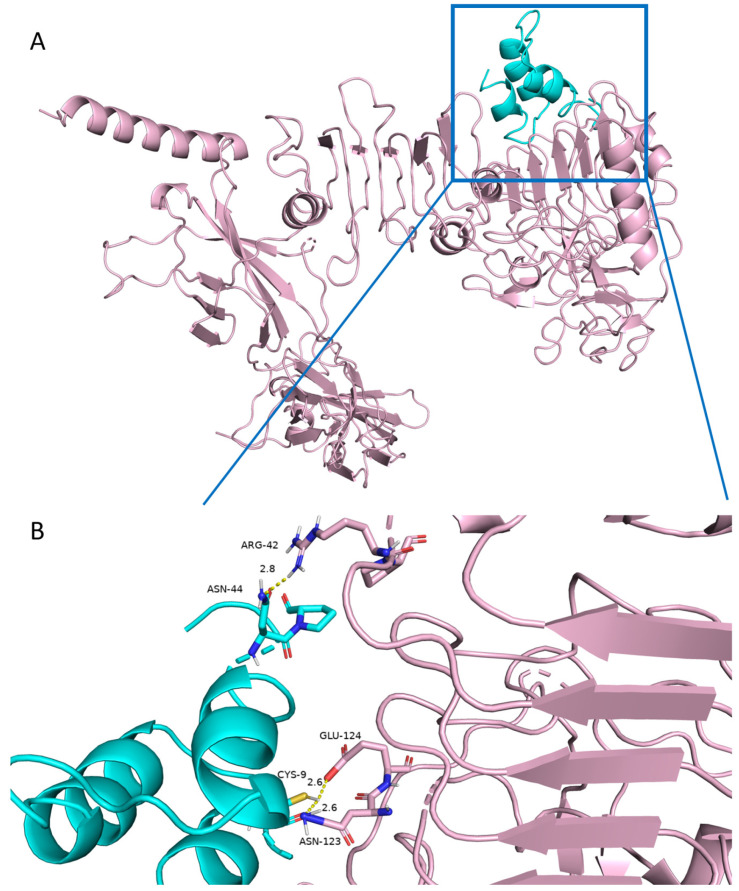
Molecular docking revealed the binding of ILP-Ap04 with human insulin receptor. (**A**) Docking pattern diagram of ILP-Ap04 (blue) and human insulin receptor (pink); (**B**) the interaction diagram of key amino acid residues at the interface between ILP-Ap04 (blue) and human insulin receptor (pink).

## Data Availability

The original data presented in the study are included in the article; further inquiries can be directed to the corresponding author.

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
