# Peer review of "Synthesis and Hypoglycemic Effect of Insulin from the Venom of Sea Anemone Exaiptasia diaphana"

_marinedrugs, 2024, doi:10.3390/md22030111_

Round 1

Reviewer 1 Report

Comments and Suggestions for Authors

The manuscript "Synthesis and Hypoglycemic Effect of Insulin from the Venom of Sea Anemone Exaiptasia diaphana" by Qiqi Guo and Tianle Tang et al. is dedicated to solid-phase peptide synthesis of transcriptome-identified insulin-like peptide. Full peptide refolding produced three products of partial disulfide formation which were subjected to circular dichroism analysis. CD-spectra deconvolution confirmed that secondary structure of a final product is close enough to human insulin. Danio rerio model of diabetes was used to demonstrate strong hypoglycemic effects of the peptide (comparable to human insulin in this model). D. rerio larvae locomotor activity was depressed by the studied peptide in a concentration-dependent manner, whereas human insulin did not show any activity. Authors also deploy rigid protein-protein docking to illustrate the plausibility of ILP interaction with human insulin receptor.
The manuscript is of a decent quality, contains high quality illustrations and is methodologically accurate. It falls into the Marine Drugs scope perfectly and I think it can be published in this journal. I have several points to address before publication.

Details of the statistical analysis are not adequately described:
1. Figure 4 legend says that unpaired t-test was used to make comparisons. However, multiple comparisons (within each time point) need to be done using either some variant of ANOVA or using corrected t-test. Please indicate exactly what type of correction was used in this case;
2. Figure 5 legend does not provide info regarding the statistical test used to assess differences. Again, multiple comparisons requires ANOVA or corrected t-test (e.g. Bonferroni).

Docking description inaccuracy:
Line 235 "π− π interaction between Pro-43 (receptor) and Pro-45 (ligand)" Proline residues absolutely can not participate in pi-pi interactions due to the absence of pi-orbitals. Proline is not an aromatic residue.

Minor English editing is required:

1. Line 37 "found to selectively blocks... It represents ..."

2. Line 50 " to unforld"

Comments on the Quality of English Language

In my view, English needs only minor spellcheck. Some phrases could be split to ease the understanding.

Author Response

Reviewer 1

The manuscript "Synthesis and Hypoglycemic Effect of Insulin from the Venom of Sea Anemone Exaiptasia diaphana" by Qiqi Guo and Tianle Tang et al. is dedicated to solid-phase peptide synthesis of transcriptome-identified insulin-like peptide. Full peptide refolding produced three products of partial disulfide formation which were subjected to circular dichroism analysis. CD-spectra deconvolution confirmed that secondary structure of a final product is close enough to human insulin. Danio rerio model of diabetes was used to demonstrate strong hypoglycemic effects of the peptide (comparable to human insulin in this model). D. rerio larvae locomotor activity was depressed by the studied peptide in a concentration-dependent manner, whereas human insulin did not show any activity. Authors also deploy rigid protein-protein docking to illustrate the plausibility of ILP interaction with human insulin receptor.

The manuscript is of a decent quality, contains high quality illustrations and is methodologically accurate. It falls into the Marine Drugs scope perfectly and I think it can be published in this journal. I have several points to address before publication.

Reply:

Thank you for your affirmation and praise of our manuscript, including the quality of the manuscript, the quality of the illustrations, and the methodology. We believe that the innovative research presented in this article has contributed valuable insights to the field of biomedical research. The synthesis of insulin from sea anemone venom has broad application prospects in diabetes. Thank you to the reviewers for their constructive comments on the details of the manuscript, and we have actively made revisions accordingly.

Details of the statistical analysis are not adequately described:

  1. Figure 4 legend says that unpaired t-test was used to make comparisons. However, multiple comparisons (within each time point) need to be done using either some variant of ANOVA or using corrected t-test. Please indicate exactly what type of correction was used in this case;
  2. Figure 5 legend does not provide info regarding the statistical test used to assess differences. Again, multiple comparisons requires ANOVA or corrected t-test (e.g. Bonferroni).

Reply:

Thank you for the suggestion. We have re-analyzed the data in Fig 4 and Fig 5 with one-way ANOVA and updated the figure legends accordingly.

Docking description inaccuracy:

Line 235 "π− π interaction between Pro-43 (receptor) and Pro-45 (ligand)" Proline residues absolutely can not participate in pi-pi interactions due to the absence of pi-orbitals. Proline is not an aromatic residue.

Reply:

Thank you to the reviewer for the constructive comments on the molecular docking section of the manuscript. We apologize for the inaccuracy in our description of the π - π interaction, acknowledging this as an oversight. After a thorough review and examination of the molecular docking results, we agree with the reviewer's viewpoint and have deleted the statement about the predicted π - π interaction from the manuscript..

Minor English editing is required:

  1. Line 37 "found to selectively blocks... It represents ..."

Reply:

Thank you for revising the details of the manuscript. We have revised this according to the reviewers' comments. ”...was found to selectively block Kv1.3 channel, and it represents...” (lines 37).

  1. Line 50 " to unforld"

Reply:

Thank you very much for pointing out the typo. We have changed unforld to unfold. (lines 51).

Comments on the Quality of English Language

In my view, English needs only minor spellcheck. Some phrases could be split to ease the understanding.

Reply:

Based on your suggestion, we have conducted a comprehensive check on the vocabulary and grammatical errors in the manuscript and have corrected the erroneous parts.

Reviewer 2 Report

Comments and Suggestions for Authors

The manuscript: “Synthesis and Hypoglycemic Effect of Insulin from the Venom of Sea Anemone Exaiptasia diaphana” are showing satisfactory procedures for the synthesis of peptide ILP-Ap04 from sea anemones. The authors study their role in potentially hypoglycemic activity by in vivo tests using zebrafish and zebrafish larvae. Furthermore, the authors characterized the molecular binding interaction of ILP-Ap04 with the human insulin receptor through docking studies and propose a robust model structure of the complex. The results were very well designed and support the conclusions. The discussion is quite sustainable and I have some suggestions for submitting the authors' evaluation. I have a general question: why haven't the authors advanced in vivo studies using mammalian models? Therefore, I would like to congratulate the authors for their work and the article is ready for publication

1. Introduction

Line 45:  The acronyms ShK and EGF should be provided in the sentense ...ShK domain ... EGF like domains[8].

Line 183:

...which provides new avenues for the development and utilization of insulin-based therapeutics.

Comment: The authors could explain how this data could be used for therapeutic purposes. Still as an introductory form to be better addressed in the discussion.

3. Discussion

The statements below draw attention to the biotechnological potential of the studied peptide. However, a more precise and in-depth analysis of what was revealed by structural and in vivo data was lacking. Could these data be better than those already discovered in other organisms or already developed as medicines? I suggest that the authors deepen the discussion on the potential biotechnological application of the relevant results of this study.

Line 250

- The identification of anti-diabetic ILPs in marine organisms such as cone snails and sea anemones exemplify this new horizon...

Line 252

- Such ILPs could represent exciting exploitable scaffolds for future drug discovery in diabetes, as well as providing tools to allow for a better understanding of cell...

Line 295

- ILP-Ap04 in modulating blood glucose levels in zebrafish models further corroborates its therapeutic potential...

Line 310

- This novel interaction suggests alternative pathways for insulin receptor activation, providing a foundation for the design of new insulin analogs that could offer therapeutic advantages over existing treatments.

Author Response

Reviewer 2

The manuscript: “Synthesis and Hypoglycemic Effect of Insulin from the Venom of Sea Anemone Exaiptasia diaphana” are showing satisfactory procedures for the synthesis of peptide ILP-Ap04 from sea anemones. The authors study their role in potentially hypoglycemic activity by in vivo tests using zebrafish and zebrafish larvae. Furthermore, the authors characterized the molecular binding interaction of ILP-Ap04 with the human insulin receptor through docking studies and propose a robust model structure of the complex. The results were very well designed and support the conclusions. The discussion is quite sustainable and I have some suggestions for submitting the authors' evaluation. I have a general question: why haven't the authors advanced in vivo studies using mammalian models? Therefore, I would like to congratulate the authors for their work and the article is ready for publication

Reply:

We express our gratitude to the reviewer for the positive feedback and recognition of our work. In this study, sea anemone insulin was synthesized using SPPS. Due to the low yield and challenges in achieving proper oxidative folding of the peptide because of its complexity, our ability to obtain sufficient peptides was limited. As mammalian models require a large amount of peptides, we embarked on a preliminary exploration using the zebrafish model, which has been demonstrated to be a reliable animal model by Peter Ahorukomeye et al. (eLife) for the study of conus insulin. Moving forwared, we will optimize the oxidative folding conditions or modify the sea anemone insulin sequence to increase the peptide yield..These adjustments will pave the way for in vivo studies on mammalian models to verify their hypoglycemic efficacy.

  1. Introduction

Line 45: The acronyms ShK and EGF should be provided in the sentense ...ShK domain ... EGF

like domains[8].

Reply:

Thanks to the reviewer for pointing this out. We have provided the full names for  the acronyms ShK and EGF:”κ1.3-SHTX-She1a (ShK), Kunitz-type, β-Defensin and epidermal growth factor-like (EGF-like)“ (lines 45).

Line 183:

...which provides new avenues for the development and utilization of insulin-based therapeutics.

Comment: The authors could explain how this data could be used for therapeutic purposes. Still as an introductory form to be better addressed in the discussion.

Reply:

Thank you very much for your suggestion. We agree with the reviewer’s advice and have modified the relevant results and discussion sections. This question is addressed together with the following questions regarding the Discussion section as below.

  1. Discussion

The statements below draw attention to the biotechnological potential of the studied peptide. However, a more precise and in-depth analysis of what was revealed by structural and in vivo data was lacking. Could these data be better than those already discovered in other organisms or already developed as medicines? I suggest that the authors deepen the discussion on the potential biotechnological application of the relevant results of this study.

Line 250

- The identification of anti-diabetic ILPs in marine organisms such as cone snails and sea

anemones exemplify this new horizon...

Line 252

- Such ILPs could represent exciting exploitable scaffolds for future drug discovery in diabetes,

as well as providing tools to allow for a better understanding of cell...

Line 295

- ILP-Ap04 in modulating blood glucose levels in zebrafish models further corroborates its

therapeutic potential...

Line 310

- This novel interaction suggests alternative pathways for insulin receptor activation, providing a

foundation for the design of new insulin analogs that could offer therapeutic advantages over

existing treatments.

Reply:

Thank you very much for your suggestion. We have modified the discussions based on your suggestions.

“When tested in zebrafish, ILP-Ap04, like human insulin, significantly lowers blood glucose in the STZ and glucose-induced model of diabetes (Figure 4). Our results are consistent with the findings of Helena Safavi Hemami et al., where human insulin at the same dosage decreased blood glucose [24,30]. Therefore, the structural and functional aspects of ILP-Ap04 could be compatible with human insulin receptors. This compatibility suggests a potential for cross-species efficacy, advocating that sea anemone-derived insulin could broaden the spectrum of insulin therapies. 

The differential efficacy of ILP-Ap04 in STZ versus glucose-induced diabetes models (Figure 4C) may provide valuable insights into the mechanisms of insulin receptor interaction and insulin resistance. Understanding how ILP-Ap04 modulates blood glucose levels could lead to the development of insulins with improved receptor affinity or reduced propensity for inducing insulin resistance. In addition, the distinct hypoglycemic effects observed in different diabetes models highlight the possibility of tailoring insulin therapies to specific types of diabetes or patient needs. Insulins like ILP-Ap04 could be optimized for longer shelf-life, reduced risk of eliciting immune responses, and for particular diabetic conditions, improving personalized medicine approaches in diabetes care” (lines 294-309).

Reviewer 3 Report

Comments and Suggestions for Authors

Dear editor,

I hope this message finds you well. I recently had the opportunity to review the article titled "Synthesis and Hypoglycemic Effect of Insulin from the Venom 2 of Sea Anemone Exaiptasia diaphana" submitted to your esteemed journal. I must commend the authors on their innovative research, which undoubtedly contributes valuable insights to the field of biomedicine. The exploration of insulin synthesis from the venom of Sea Anemone Exaiptasia diaphana is both intriguing and promising for potential applications in diabetes management.

While the article presents groundbreaking results, I would like to suggest a few minor adjustments to enhance its overall quality. Overall, I believe this article has the potential to make a significant impact in the biomedical community.

Specific comments.

Line 50. Change unforld by unfold

Line 52: Scientific name in italics

Line 69 to 72: Authors indicate that the current study is based on one of the fifth insulin peptides previously described, however, there is not information regarding why the authors select the o4 insuline peptide, please include information on the introduction.

Figure 1. It is a very informative figure, however, it will be good to include an overlapping image of the human insulin and the ILP-Ap04

Line 143. Figure

Section 2.6 Docking, it would be very helpful to do a overlapping of the docking of INSR and  human insulin and ILP-Ap04 to differentiate the main changes, also related to the lack of FFY motif.

Line 286. Comparison

Line 294. Ap04 is

Authors indicate that the FFY motif is implicated in the high affinity by the receptors, how does it work without this motif, and what are the implications of this main difference, is there any advantage (evolutive).

How is the affinity of IlO1_i1, ILP-Ap04 compared to human insulin receptor, compared to human insulin with FFY motif?

Comments on the Quality of English Language

There are some typos in the manuscript, it requires double checking

Author Response

Reviewer 3

Dear editor,

I hope this message finds you well. I recently had the opportunity to review the article titled "Synthesis and Hypoglycemic Effect of Insulin from the Venom 2 of Sea Anemone Exaiptasia diaphana" submitted to your esteemed journal. I must commend the authors on their innovative research, which undoubtedly contributes valuable insights to the field of biomedicine. The exploration of insulin synthesis from the venom of Sea Anemone Exaiptasia diaphana is both intriguing and promising for potential applications in diabetes management.

While the article presents groundbreaking results, I would like to suggest a few minor adjustments to enhance its overall quality. Overall, I believe this article has the potential to make a significant impact in the biomedical community.

Specific comments.

Line 50. Change unforld by unfold

Reply:

Thank you very much for pointing out the error in this word. We have changed unforld to unfold. (lines 51).

Line 52: Scientific name in italics

Reply:

We appreciate the reviewer’s suggestion and have written all the species names in italics such as ”Drosophila melanogaster. (lines 52).

Line 69 to 72: Authors indicate that the current study is based on one of the fifth insulin peptides previously described, however, there is not information regarding why the authors select the o4 insuline peptide, please include information on the introduction.

Reply:

Thank you for suggestion. Following this advice, we have added now explained why this particular peptide was selected. ”To further understand the biological functions of these sea anemone ILPs, this study focuses on ILP-Ap04, one of the five insulin sequences identified in Exaiptasia diaphana. Distinct from its counterparts, ILP-Ap04 exhibits significant specificity in its amino acid sequences and 3D structures. Therefore, it was synthesized through solid-phase peptide synthesis (SPPS) to investigate its hypoglycemic activity in zebrafish models of hyperglycemia and behavioral assays.” (lines 68-73).

Figure 1. It is a very informative figure, however, it will be good to include an overlapping image of the human insulin and the ILP-Ap04

Reply:

We appreciate this suggestion. We tried to overlay these structures for a combined display, but the outcome was unsatisfactory. Therefore, we prefer to present and compare them individually.

Line 143. Figure

Reply:

Thanks to the reviewer for the reminder, we have changed the figure to Figure.

Section 2.6 Docking, it would be very helpful to do a overlapping of the docking of INSR and  human insulin and ILP-Ap04 to differentiate the main changes, also related to the lack of FFY motif.

Reply:

We thank the reviewer for this suggestion.  Similar to the case with Figure 1, we attempted to overlay these structures for a combined display, but the outcome was unsatisfactory.  It seems challenging to overlay human and conus insulin structures without losing clarity, as demonstrated in the following paper.

(Ahorukomeye, P.; Disotuar, M.M.; Gajewiak, J.; Karanth, S.; Watkins, M.; Robinson, S.D.; Flórez Salcedo, P.; Smith, N.A.; Smith, B.J.; Schlegel, A.; et al. Fish-hunting cone snail venoms are a rich source of minimized ligands of the vertebrate insulin receptor. Elife 2019, 8. https://doi.org/10.7554/eLife.41574.)

Line 286. Comparison

Reply:

Thank you for your careful proofreading of the original manuscript. We have modified this one。

Line 294. Ap04 is

Reply:

Thanks for catching this error. This was revised in the revised manuscript:”ILP-Ap04 is……”

Authors indicate that the FFY motif is implicated in the high affinity by the receptors, how does it work without this motif, and what are the implications of this main difference, is there any advantage (evolutive).

Reply:

Thank you for your questions and suggestions. In 2021, Michela L Mitchell et al. published an article on sea anemone insulin and mentioned ”The IlO1_i1 peptide lacks the key FFY motif at the end of the B-chain, which in human insulin engages with the high-affinity binding site on the IR. Cone snail insulin Con-Ins G1 also lacks this motif and yet binds with reasonable affinity.”The sea anemone insulin found in this study also lacks the FFY motif, but it has been confirmed to have hypoglycemic activity through zebrafish experiments. Through molecular docking, key amino acids in human insulin receptors and Arg52, Ser53, Arg68, and Arg69 in ILP Ap04 were identified, promoting strong hydrophobic interactions at their interfaces. Moreover, ILP-Ap04 engages the pocket of human insulin receptor through three hydrogen bonds (Figure 6B), involving Arg-42 (2.8 Å), Glu-124 (2.6 Å), and Asn-123 (2.6 Å). These residues may play an important role in maintaining the structural stability of the receptor-ILP-Ap04 complex. Therefore, we speculate that the same mechanism of action as insulin provides a new strategy for hypoglycemic drug design.

How is the affinity of IlO1_i1, ILP-Ap04 compared to human insulin receptor, compared to human insulin with FFY motif?

Reply:

The precise mechanism of action distinguishing insulin variants lacking the FFY motif and human insulin requires further in-depth research, which could inspire new avenues for the development of hypoglycemic drugs in the future. Nevertheless, the following literature discusses the relationship between insulin receptor affinity and the FFY motif, guiding our initial exploration into this area.

In 2021, Michela L Mitchell et al. revealed: The IlO1_i1 peptide lacks the key FFY motif at the end of the B-chain, which in human insulin engages with the high-affinity binding site on the IR. Cone snail insulin Con-Ins G1 also lacks this motif and yet binds with reasonable affinity. Structural and analogue studies of Con Ins G1 revealed that two tyrosine residues at positions B15 and B20 substitute for the lack of the FFY motif. Despite the IlO1_i1 peptide having a tyrosine at B15, it is unable to bind to the IR, suggesting that other sequence differences prevent binding. (doi: 10.3390/biom11121785).

In 2020, Xiaochun Xiong et al. revealed: Con-Ins G1 is an outlier among known natural insulin-like molecules, having four posttranslational modifications that enhance its interaction with hIR and lacking the C-terminal residues of the canonical insulin B chain. The relatively high potency of Con-Ins G1 and several other cone snail insulin-like peptides28 with respect to hIR is unexpected, as the Bchain C-terminal octapeptide (inclusive, especially, of the aromatic triplet PheB24-PheB25-TyrB26) is critical for high-affinity binding of human insulin to hIR. (DOI: 10.1038/s41594-020-0430-8).

Their work has been discussed in this manuscript.

Comments on the Quality of English Language

There are some typos in the manuscript, it requires double checking

Reply:

We thank the reviewer for this suggestion. We have conducted a comprehensive check on the vocabulary and grammar errors in the manuscript and have corrected the erroneous parts.

Finally, thanks again to the editor and reviewers for your positive feedback and insightful suggestions for improving our manuscript. We appreciate all the help.